# Ferroelectric compute-in-memory annealer for combinatorial optimization problems

Xunzhao Yin [1,2,6], Yu Qian [1,6], Alptekin Vardar[3], Marcel Günther[3], Franz Müller [3], Nellie Laleni[3], Zijian Zhao[4], Zhouhang Jiang[4], Zhiguo Shi[1,2], Yiyu Shi [4], Xiao Gong [5], Cheng Zhuo [1,2,7] ✉, Thomas Kämpfe [3,7] ✉ & Kai Ni [4,7] ✉

Computationally hard combinatorial optimization problems (COPs) are ubiquitous in many applications. Various digital annealers, dynamical Ising machines, and quantum/photonic systems have been developed for solving COPs, but they still suffer from the memory access issue, scalability, restricted applicability to certain types of COPs, and VLSI-incompatibility, respectively. Here we report a ferroelectric field effect transistor (FeFET) based compute-in-memory (CiM) annealer for solving larger-scale COPs efficiently. Our CiM annealer converts COPs into quadratic unconstrained binary optimization (QUBO) formulations, and uniquely accelerates in-situ the core vector-matrix-vector (VMV) multiplication operations of QUBO formulations in a single step. Specifically, the three-terminal FeFET structure allows for lossless compression of the stored QUBO matrix, achieving a remarkably 75% chip size saving when solving Max-Cut problems. A multi-epoch simulated annealing (MESA) algorithm is proposed for efficient annealing, achieving up to 27% better solution and ~2X speedup than conventional simulated annealing. Experimental validation is performed using the first integrated FeFET chip on 28nm HKMG CMOS technology, indicating great promise of FeFET CiM array in solving general COPs.

Combinatorial optimization problems (COPs), as shown in Fig. 1a, are prevalent in diverse fields, including logistics, resource allocation, communication network design, finance, drug discovery, and transportation systems, etc.[1–4]. Often, these problems belong to the class of non-deterministic polynomial-time-hard (NP-hard) problems, representing some of the most challenging computational tasks in the NP domain. Solving COPs using digital computers based on the von Neumann architecture poses difficulties, given the exponential growth in required resources regarding the computational power and latency as the problems scale up[5–7]. Therefore, there is a pressing need to explore novel hardware design with alternative architectures and algorithms that can efficiently tackle COPs. This research frontier holds crucial implications for real-world applications, with the potential to address complex and resource-intensive problems with greater effectiveness.

Many COPs, including graph coloring, Max-Cut, and traveling salesman problem, etc., can be mapped to the Ising spin glass model or often go by the name QUBO (i.e., quadratic unconstrained binary optimization)[8], which have emerged as a powerful framework for effectively modeling and solving a wide range of COPs[9]. In this framework, the problem variables are elegantly represented as spins, and the interactions or constraints between variables are represented as spin-to-spin couplings. The objective function of the problem can then be mapped to the Hamiltonian energy function of the Ising model. The

[1]Zhejiang University, Hangzhou, China. [2]Key Laboratory of CS&AUS of Zhejiang Province, Hangzhou, China. [3]Fraunhofer IPMS, Dresden, Germany. [4]University of Notre Dame, Notre Dame, USA. [5]National University of Singapore, Singapore, Singapore. [6]These authors contributed equally: Xunzhao Yin, Yu Qian. [7]These authors jointly supervised this work: Cheng Zhuo, Thomas Kämpfe, Kai Ni. ✉e-mail: czhuo@zju.edu.cn; thomas.kaempfe@ipms.fraunhofer.de; kni@nd.edu

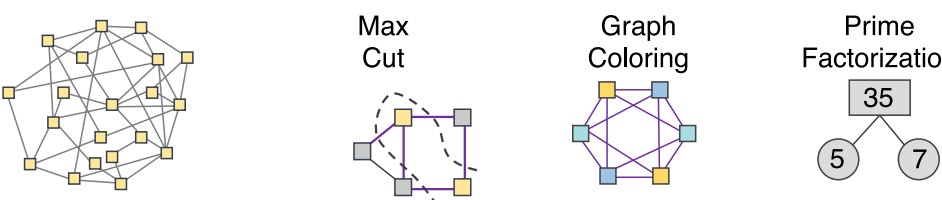

**a    Combinatorial optimization problems**

Max Cut

Graph Coloring

Prime Factorization

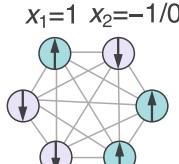

$x_1 = 1$  $x_2 = -1/0$

**b    Quadratic Unconstrained Binary Optimization**

minimize/maximize:  $f = \mathbf{x}^T Q \mathbf{x}$

$Q$: a matrix, mapping a problem
$\mathbf{x}$: binary state vector

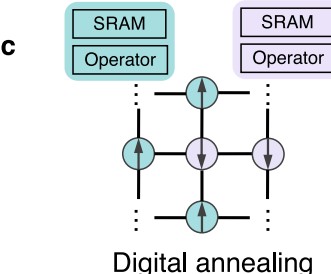

**c**  Digital annealing

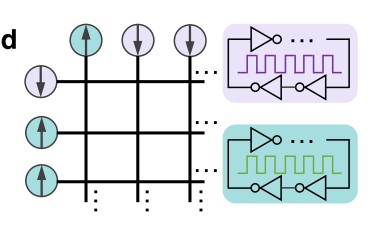

**d**  Dynamic system

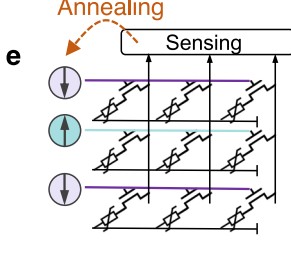

Annealing

**e**  CiM system

**f        QUBO formulation**

**FeFET crossbar array**

Objective function computed with CiM array

$$f = \mathbf{x}^T Q \mathbf{x} \propto \sum_i I_i$$

$$\mathbf{x}^T = (x_1\ x_2\ \dots\ x_N)$$

$$x_i \in \{0,1\}$$

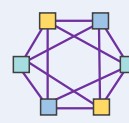

$Size(Q') \leq Size(Q)$   ✂  **Model compression via general QUBO form with asymmetric input vectors**

**g        QUBO with asymmetric input vectors**

**Compressed crossbar array**

Objective function computed with compressed CiM array

$$f = \mathbf{x}^T Q' \mathbf{y} \propto \sum_i I_i$$

$$\mathbf{x}^T = (x_1\ x_2\ \dots\ x_K)\quad \mathbf{y} = (y_1\ y_2\ \dots\ y_M)^T\quad x_i, y_i \in \{0,1\}$$

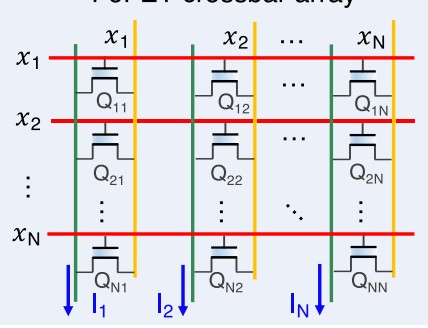

**Fig. 1 | Acceleration of solving COPs with CiM array. a** COPs (e.g., Max-Cut, graph coloring, prime factorization etc.) can be converted to **b** $\mathbf{x}^T Q \mathbf{x}$ QUBO formulation. Many hardware systems, including **c** digital annealing system, **d** dynamical system, and **e** CiM system are promising for solving COPs. **f** Due to the unique characteristics of FeFET-based CiM crossbar, it can implement typical $\mathbf{x}^T Q \mathbf{x}$ QUBO formulation with symmetrical input vectors. **g** The array can also accelerate a more general and compact $\mathbf{x}^T Q' \mathbf{y}$ QUBO formulation with asymmetrical vectors, thus achieving high efficiency and low cost.

solution of a problem then corresponds to the combination of spins that minimizes the Ising Hamiltonian $H_P$, which can be formulated as follows:

$$\min H_P = \sum_{i,j=1}^{N} J_{ij}\sigma_i\sigma_j + \sum_{i=1}^{N} h_i\sigma_i \qquad (1)$$

where $N$ denotes the number of spins, and $\sigma_i \in \{1, -1\}$ represents the state of spin $i$. $J_{ij}$ and $h_i$ stand for the coupling between spin $i$ and $j$ and the self-coupling of spin $i$, respectively. Through a simple variable change $\sigma_i = 1 - 2x_i$, $x_i \in \{0, 1\}$, the Ising Hamiltonian (Eq. (1)) can be readily transformed into a QUBO matrix form[10,11] as

$$H_{QUBO} = \mathbf{x}^T Q \mathbf{x} \qquad (2)$$

where $\mathbf{x} = (x_1, x_2, ..., x_n)$, and $Q$ is a symmetric or an equivalent upper triangular $n \times n$ matrix[12,13], as shown in Fig. 1b. For instance, consider a Max-Cut problem defined on an undirected graph G(V, E), where $V$ represents the set of vertices, and $E$ represents the set of edges[12]. The Max-Cut problem is mapped into the QUBO form by introducing a binary variable $x_i \in \{0, 1\}$ for each vertex $i \in V$, where $x_i$ takes the value 1 if vertex $i$ is assigned to one set and 0 if it belongs to the other set. The objective function of the Max-Cut problem can then be formulated as follows:

$$\min \sum_{(i,j)\in E} (2x_i x_j - x_i - x_j) \qquad (3)$$

Considering the binary $x_i$, such a form can be easily represented as the $\mathbf{x}^T Q \mathbf{x}$ QUBO form[12,13]. The conversion of other COPs, such as graph coloring problems and prime factorization problems, into the QUBO form is further elaborated in Sec. 1 of Supplementary Information.

To solve COPs efficiently, various alternative computing hardware are under active research. Figure 1c, d briefly summarizes different electronic implementations. One class of hardware are digital ASIC annealers, where various annealing algorithms are implemented in digital circuits[14–17]. Usually the spin coupling matrix is stored in memory and data need to be frequently transferred between memory and computing units for energy computation and annealing, which can be energy- and time-consuming as the problem scales up. An attractive alternative is dynamical system Ising machines, where the intrinsic system dynamics and tendency to settle at lowest energy state is exploited to solve the COPs, as shown in Fig. 1d. Once the spin coupling matrix are programmed within the hardware, these solvers naturally explore the solution space and ultimately find the spin combination that minimizes the Ising energy without explicitly executing annealing algorithms. Examples include the oscillator-based Ising machine (OIM)[18–20], latch-based Ising machine[21–23], and optical-based coherent ising machine (CoIM)[24–28].

While the concept of such a system holds immense promise, there are several challenges that remain to be addressed. First, the dynamics and robustness of dynamical Ising solvers is highly sensitive to the coupling implementations between spins, as a slight deviation in coupling strength can lead to convergence disruption of the solution[19,22]. Therefore, it poses a significant challenge in precisely mapping the spin coupling matrix into hardware. Second, exploiting dynamical Ising solvers to their full potential requires mapping the entire problem onto a single solver. For large scale problems that are beyond the capacity of the solver, how to efficiently map the problems to multiple separate chips and implement chip-to-chip communication while maintaining system dynamics requires substantial work. Therefore, scaling of dynamical Ising solver is a critical challenge. Lastly, integration of self-interaction into these dynamical Ising solvers is not straightforward, thus allowing easy mapping of only a subclass of COPs without self-interaction terms, such as Max-Cut, Sherrington-

Kirkpatrick models, etc[18,29]. Many COPs requiring self-interaction terms after mapping to the Ising model, including graph coloring, prime factorization, bin packing, etc., remain yet to be solved by dynamical Ising solvers. Other unconventional approaches, including quantum and photonic implementations, generally utilize their unique physical behavior to directly represent the Ising models. However, many of them are challenging to integrate into silicon VLSI technologies. For example, the D-Wave quantum annealers proposed in refs. 30–32 require expensive cryogenic cooling and exhibit limited connectivity between spins. Optical Ising machine consumes extremely long optic fiber to implement the solver, making its integration highly challenging[25].

In this article, we perform a hardware-algorithm co-design of a compute-in-memory (CiM) based annealer to efficiently solve QUBO formulations, thus the COPs, as shown in Fig. 1e. The most well-known CiM hardware system is probably the crossbar array for acceleration of the vector-matrix multiplication (VMM), a core operation in neural networks[33]. In this scheme, the matrix is stored in memory, including volatile and nonvolatile memory (NVM), and the VMM computations are performed in CiM arrays without energy-consuming and slow data movement between memory and computing units, thus exhibiting superior energy efficiency. Drawing inspiration from this, and recognizing that the QUBO formulation is composed of a vector-matrix-vector (VMV) multiplication as shown in Eq. (2), this article aims at expediting the in-situ VMV multiplication through CiM approach, thus accelerating solving COPs. Our CiM annealer could potentially address the aforementioned challenges faced by digital annealers and dynamical Ising solvers, offering several advantages: (i) our CiM approach stores the QUBO matrix in memory and directly performs VMV multiplication in memory, avoiding the data movement bottleneck seen in digital annealers; (ii) by programming multiple FeFET devices with binary states to represent a single matrix coefficient and performing the VMV multiplication in analog domain, CiM annealer is intrinsically robust against the noise and inaccuracy of the coupling matrix mapping. In contrast, Ising solvers can be vulnerable to these issues; (iii) Unlike dynamic Ising solvers, which rely on the overall system dynamics to solve COPs, our CiM-based approach easily handles larger-scale problems beyond the capacity of our chip by decomposing the corresponding QUBO formulation into smaller forms, then independently mapping and computing these forms across multiple CiM chips; (iv) Lastly, our CiM array can readily implement self-interaction terms within the QUBO formulation by programming the diagonal matrix coefficient value onto the crossbar cells. In conclusion, CiM approach when seamlessly integrated with efficient annealing algorithms, could offer a powerful hardware platform for COPs.

Here we propose to develop an ferroelectric field effect transistor (FeFET) based CiM crossbar array to accelerate VMV multiplications of QUBO, as shown in Fig. 1f. FeFETs based on ferroelectric HfO$_2$ are a prime candidate technology platform to implement CiM system for in-situ VMV multiplication. First, it is naturally a three-terminal nonvolatile device, ideal for VMV multiplication, where the coupling matrix element can be stored in the polarization state of the FeFET and the two inputs (not necessarily identical) can be applied on the gate and drain, respectively. On the contrary, other two-terminal NVM based CiM system would require an VMM operation to calculate the intermediate result, and then apply another dot multiplication in digital domain to complete the VMV multiplication. Second, HfO$_2$ based FeFET exhibits superior energy efficiency with its electric field driven polarization switching mechanism and high ON/OFF ratio[34,35], while current-driven memristor devices require additional access transistors and complex sensing circuitry, leading to much more energy consumption than FeFETs. Third, FeFETs stand out due to its CMOS compatibility and scalability[34,35], while embedded flash struggles to scale beyond the 28nm node[36]. When performing VMV multiplication, a FeFET-based CiM array necessitates lower write/read

voltages ($V_{write}/V_{read} = 4/1V$) and less write time (~$10ns$) compared to flash ($V_{write}/V_{read} = 15/4.5V$, $t_{write} = 1ms$)[37]. This results in reduced energy consumption and execution time. Therefore, a compact single FeFET CiM array is developed in this work for the QUBO computations. Compared to memristor-based Max-Cut problem solver[38,39], our work represents a significant advancement in CiM based annealers. The innovation of this work lies in: i) first proposal of a compact and effective 1FeFET1R CiM implementation for in-situ VMV multiplication by exploiting the three-terminal structure and nonvolatile storage of FeFETs. These voltage-driven devices feature with superior write energy and unique single-step 3-input multiplication capability; ii) proposing a lossless compression method for the QUBO formulation by capitalizing on the FeFET CiM array's capability to accommodate in-situ VMV multiplication with asymmetrical (non-identical) input vectors as shown in Fig. 1g, thus significantly reducing the array size of the crossbar and expanding the problem-solving capacity to larger scales; iii) introducing a multi-epoch simulated annealing (MESA) algorithm to enhance the annealing process and improve the solution quality, which can quickly find the optimal solution of COPs via iterative QUBO computations; iv) first experimental demonstration of a FeFET CiM array to showcase its efficacy in accelerating QUBO computations and highly competitive performance against other hardware alternatives in solving complex COPs. The overall working flow of our CiM based annealer is depicted in Sec. 2 of Supplementary Information and as follows: (i) A COP is initially converted into a QUBO formulation $\mathbf{x}^T Q \mathbf{x}$ as shown in Fig. 1f. (ii) This QUBO formulation is then losslessly compressed into a more general and compact form with asymmetric variable vectors $\mathbf{x}^T Q \mathbf{y}$, as shown in Fig. 1g. (iii) The QUBO matrix $Q'$ of the compressed formulation is mapped onto a FeFET-based crossbar array, which inherently performs single-step VMV multiplication. The summed current of the crossbar represents the value of the compressed QUBO objective function. (iv) The solving process utilizes a MESA algorithm. In each iteration of the annealing process, the FeFET-based array computes the QUBO formulation value, and the objective function value is determined. (v) After the MESA process, the variable vector configurations that correspond to the optimal objective function value are obtained and translated into the solution for the given COP.

## Results

### 1FeFET1R based CiM architecture

Considering the great promise of FeFET crossbar array in accelerating VMV multiplication with both symmetric and asymmetric input vectors for COPs in QUBO formulation, FeFET CiM array is designed and experimentally demonstrated. Figure 2 shows the cell and array design and experimental data illustrating the CiM hardware. The FeFET CiM chip is integrated onto an industrial 28nm high-$\kappa$ metal gate FeFET technology platform[40]. The device features an approximately 8nm doped $HfO_2$ as the ferroelectric layer, as shown in the schematic cross-section and transmission electron microscopy (TEM) cross-section in Fig. 2a. The structural similarity of FeFET to standard logic transistor, coupled with its CMOS compatibility and ultra-scalable nature of ferroelectric $HfO_2$, enables the integration of FeFETs with Si CMOS, which is leveraged in this work. For the demonstration, an 32 × 32 FeFET array is designed, where the chip layout is composed of array core, the word line (WL) driver, source line (SL)/data line (DL) driver, and the analog-to-digital converter (ADC) is shown in Fig. 2b. The fabricated chip micrograph is shown in Fig. 2c.

As shown in Fig. 2d, our approach encodes the coupling matrix element $q$ into the polarization states of the FeFET. By applying inputs $x$ and $y$ to the FeFET's gate and drain, respectively, the resulting channel current $i_{DL}$ corresponds to the scalar product of these three, i.e., $i_{DL} = x \times q \times y$. Consequently, the core computation within VMV multiplication can be implemented with minimal overhead. This sets our approach apart from other two-terminal NVM devices like memristors, which are limited to singular multiplications between the input and the stored values[41,42]. Figure 2e further shows the relationship between cell current and gate voltage (i.e., $V_{WL}$) for two memory states across 60 distinct devices. The coupling matrix element is encoded as the polarization states, programmed via +4V/-4V, $1\mu s$ gate pulses, which induce the polarization to orient towards the channel/gate-metal, and hence set the threshold voltage ($V_{TH}$) of FeFET into the low-$V_{TH}$ (i.e., $q = 1$)/high-$V_{TH}$ state (i.e., $q = 0$), respectively. By choosing an appropriate read gate bias (i.e., input $x$), the resultant cell current realizes the scalar product.

While the design is compact and elegant, a potential challenge arises from the need to manage FeFET variation, which can lead to compromised accuracy in VMV multiplications. Despite ongoing improvements in materials and processes[43], FeFET variation remains a significant factor in CiM applications, as indicated in Fig. 2f. In this work, we employ an 1FeFET1R cell structure as depicted in Fig. 2g to effectively mitigate the device variations and enhance the accuracy of the VMV multiplication. By incorporating a series resistor, the cell's ON current, regulated by the current limiter, becomes independent of the FeFET's ON current[44,45]. Such structure ensures that the presence of variation in $V_{TH}$ does not manifest as variation in the cell's ON current. As a proof of concept, each FeFET is connected with a series resistor for the same group of 60 devices. Figure 2h shows the 1FeFET1R cell $I$-$V$ characteristics, which exhibits the same $V_{TH}$ distribution as that in Fig. 2e, while its ON current variation can be significantly suppressed, as illustrated in Fig. 2i. While there is a trade-off involving the reduction in the cell's ON current, the ON/OFF ratio still exceeds 1000, ensuring that there should be no constraints on the practical array size. As a result, the 1FeFET1R CiM array is designed as illustrated in Fig. 2j, where the resistor is implemented with a fully integrated MOSFET. A more detailed description of our chip measurement can be referred to Fig. S3 in Sec. 3 of Supplementary Information. Additionally, to successfully program the array while suppressing the program disturb, a standard $V_W/3$ is adopted[46], as illustrated in Fig. S5a. Also the memory array error rate shown in Fig. S5b as a function of the write conditions clearly demonstrates that it is possible to reduce the write voltage if a long write latency can be tolerated[47]. Furthermore, the current within a column exhibits a linear relationship with the number of activated cells, corroborated across 20 different arrays as shown in Fig. 2k. Therefore, it validates the linearity and functionality of the crossbar array, and also demonstrates the tightly controlled distribution of the output current. Moreover, the CiM results are stable up to $10^5$ seconds without noticeable degradation, as shown in Fig. S5c. These results lay a robust foundation for the acceleration of VMV multiplication in this work.

Lastly, we present the mapping of the generalized QUBO form, $\mathbf{x}^T Q \mathbf{y}$, onto the 1FeFET1R CiM array, as illustrated in Fig. 2l and m. The details of the general QUBO form are depicted in Fig. 2l. The input vectors, $\mathbf{x}$ and $\mathbf{y}$, are mapped to the WL and SL inputs, respectively, as shown in Fig. 2m. This figure further shows the circuit implementation of the FeFET-based crossbar array along with its associated peripheral circuits. The QUBO matrix is mapped onto the FeFET crossbar by storing $M$-bit precision matrix elements within $M$ 1FeFET1R cells. Each cell stores a single bit of the matrix element, therefore an $n \times n$ QUBO matrix corresponds to the implementation of $n \times N$ cells, where $N$ is $n \times M$. To perform the VMV multiplication, the WL driver activates all rows of the FeFET crossbar, and the SLs of the columns are activated per the input $\mathbf{y}$ (i.e., '1' indicates ON, and '0' indicates OFF). The consecutive column outputs are directed to the column-shared analog-digital converters (ADCs), converted to digits, and further processed through Shift and Add units, generating the dot product between the stored multi-bit coupling vector and input vector $\mathbf{x}$. The final value of QUBO $\mathbf{x}^T Q \mathbf{y}$ function is then accumulated as the output of the current iteration in the annealing process and stored in the output buffer. In this way, our proposed CiM crossbar realizes the VMV multiplication

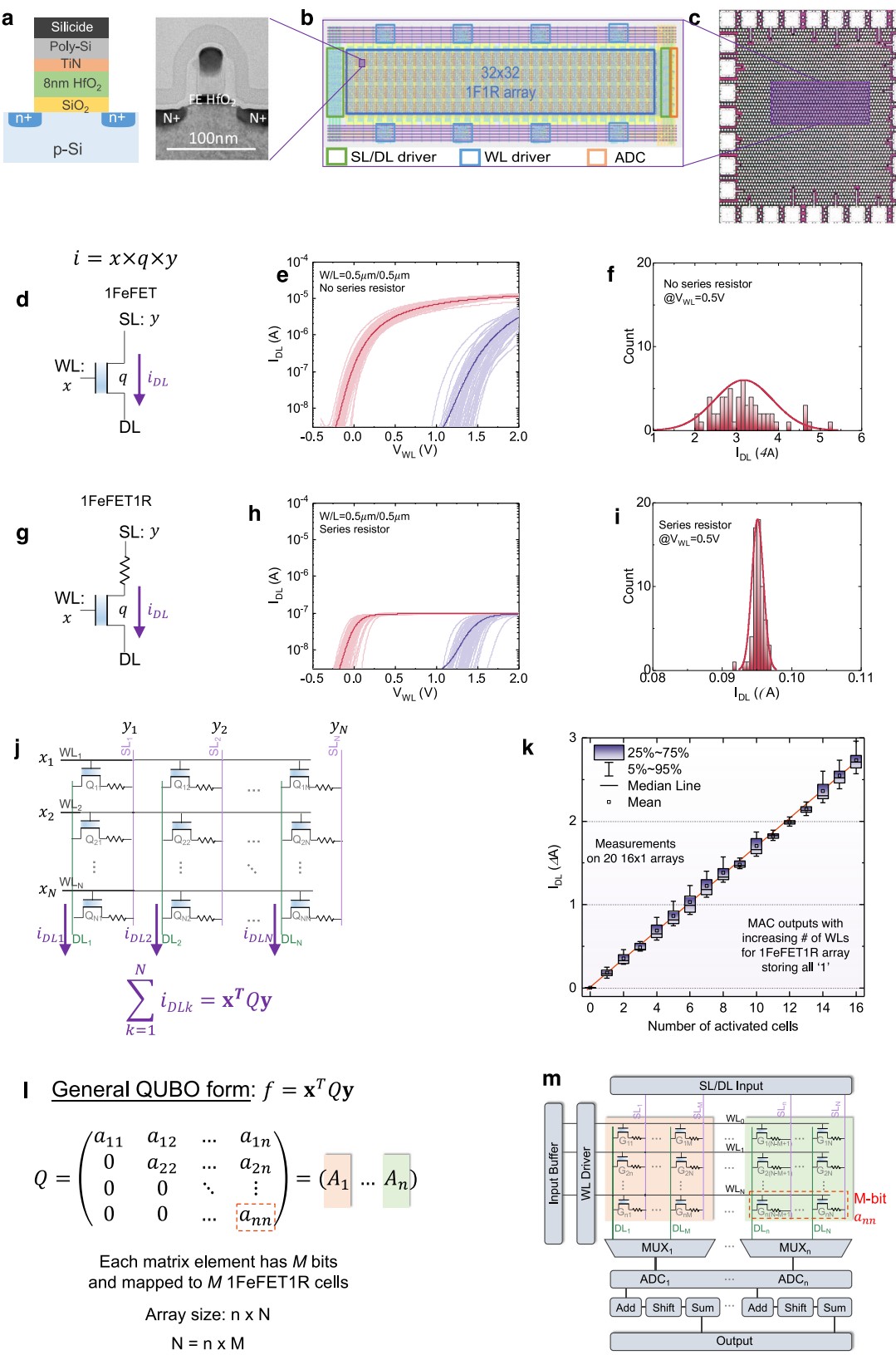

**Fig. 2 | FeFET-based CiM array for QUBO acceleration. a** FeFET schematic and TEM cross section, featuring an 8nm doped HfO$_2$ s the ferroelectric. **b** Layout of an 32 × 32 FeFET array composed of core and peripherals. **c** Micro-graph of the fabricated chip, where bond pads are visible. **d** The current of a FeFET $i_{DL}$ corresponds to the scalar product of stored value $q$ (i.e., threshold voltage $V_{TH}$), and inputs $x$ and $y$, applied at the gate and drain, respectively. **e** The $I_{DL}$-$V_{WL}$ characteristics of 60 FeFETs for the two memory states. **f** Significant ON current variation of FeFETs will result in compromised accuracy in VMV multiplications. **g** An 1FeFET1R cell

structure can suppress the $I_{ON}$ variability. **h** $I_{DL}$-$V_{WL}$ of 60 1FeFET1R cells with 1MΩ resistor. **i** A significantly narrower $I_{ON}$ distribution for the 1FeFET1R cells, thus substantially enlarging the practical CiM array size. **j** As a result, the 1FeFET1R CiM array implementing a general QUBO formulation is proposed. **k** Measured column current shows a good linearity with respect to the number of activated cells in the column, thus promising for VMV multiplication. **l** QUBO formulation mapping to **m** FeFET-based CiM array architecture.

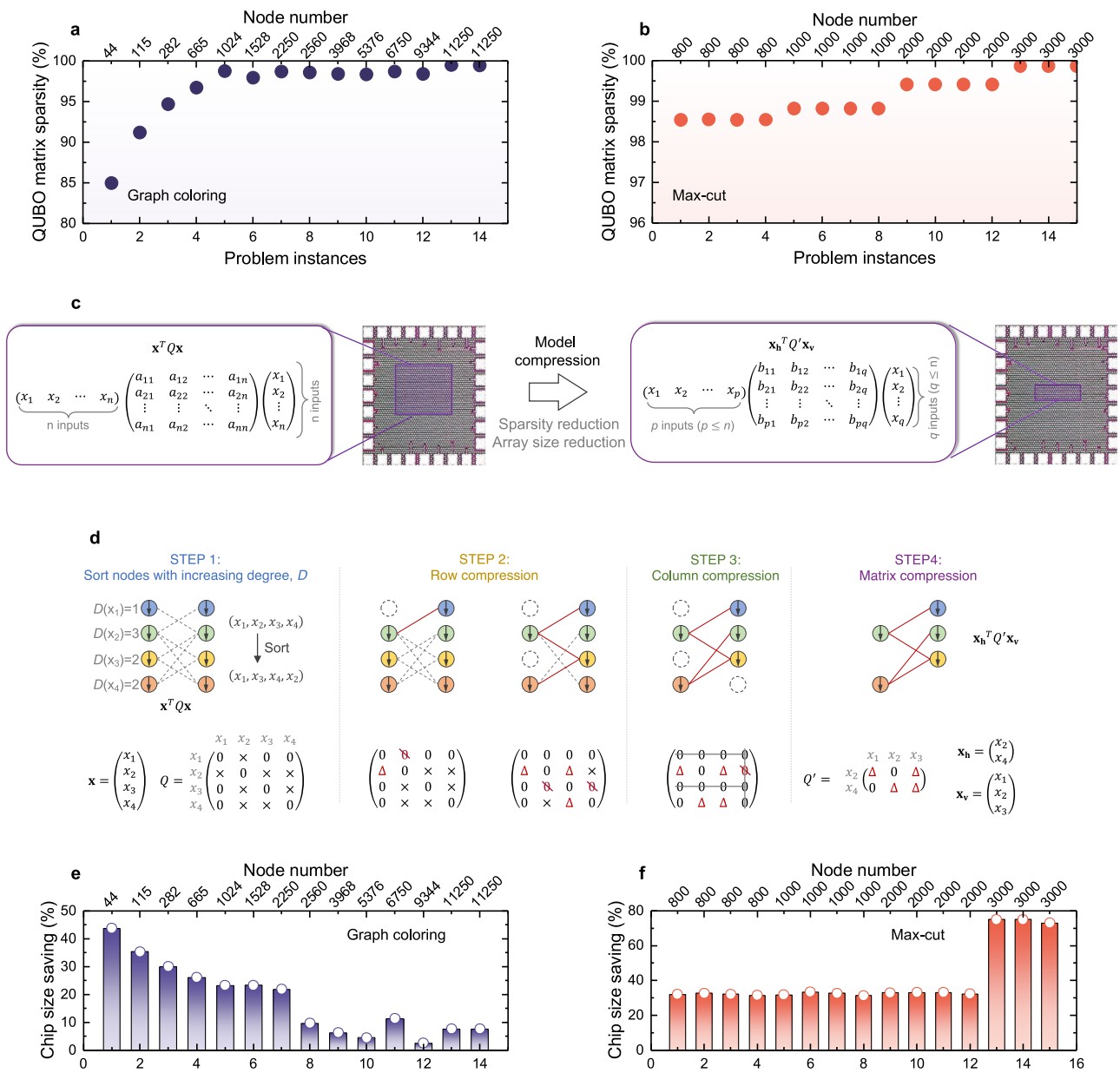

**Fig. 3 | Proposed lossless compression of the QUBO matrix.** Conventional $\mathbf{x}^T Q \mathbf{x}$ QUBO formulation for many COPs like **a** graph coloring and **b** Max-Cut is highly sparse. When directly mapped to a CiM array, significant portion of hardware could be wasted. **c** A lossless compression method is proposed to convert a large scale $\mathbf{x}^T Q \mathbf{x}$ formulation to a more compact and dense $\mathbf{x_h}^T Q' \mathbf{x_v}$ formulation by leveraging the three-terminal FeFET-based CiM array, where $\mathbf{x} = \mathbf{x_h} \cup \mathbf{x_v}$. **d** The conceptual flow of $\mathbf{x}^T Q \mathbf{x}$ compression leveraging the symmetry of the $Q$ matrix. Chip size savings after compression for **e** graph coloring and **f** Max-Cut problems, are presented in (**a**) and (**b**), respectively.

directly by simply applying two input vectors within each iteration, and ultimately solves the QUBO formulation in Eq. (2), synergizing with the efficient annealing algorithms.

## QUBO matrix lossless compression

Mapping the QUBO formulation directly onto the FeFET crossbar CiM array with two identical or symmetrical input vectors applied on the WLs and SLs, respectively, has revealed a challenge in terms of low chip utilization. This issue stems from the inherent sparsity often observed in QUBO matrix converted from COPs. Figure 3a, b show the sparsity of QUBO matrix for graph coloring problems[48] and Max-Cut problems[49], respectively, across varying problem instances with different node counts. Remarkably, the majority of the matrix elements (typically exceeding 85%) assume zero values. Therefore, when directly mapping

the QUBO matrix onto the FeFET crossbar array, a large portion of FeFET cells witin the array are programmed to state '0' (i.e., high-$V_{\text{TH}}$ state). Although these 'OFF' cells do not actively participate in the VMV multiplication during QUBO computation, they still incur additional hardware area overhead and contribute to leakage power consumption. With the expansion problem complexity and scale, the hardware size essential for accommodating the converted QUBO matrix exhibits quadratic growth with the node count, thus leading to substantial hardware resources waste. Such low hardware utilization therefore introduces formidable obstacles to CiM annealers in solving larger-scale COPs efficiently.

To minimize the hardware inefficiencies stemming from sparse matrix mapping, a lossless compression technique is proposed here, as illustrated in Fig. 3c. This approach entails pruning the sparse and

symmetric QUBO matrix, originally of size $n \times n$ within the $\mathbf{x}^T Q \mathbf{x}$ formulation, into a more compact dense matrix of size $p \times q$ within the $\mathbf{x_h}^T Q' \mathbf{x_v}$ formulation, where $\mathbf{x_h} \cup \mathbf{x_v} = \mathbf{x}$ and $p, q \leq n$. This innovative technique achieves substantial chip size reduction by capitalizing on the distinctive attributes of the three-terminal FeFET crossbar CiM array, particularly when implementing the $\mathbf{x}^T Q \mathbf{y}$ formulation, where $\mathbf{x}$ and $\mathbf{y}$ need not be symmetrical. Figure 3d illustrates the methodology of QUBO matrix compression, elucidated through a concrete example, which consists of 4 steps:

**Step 1.** The input vectors are organized in order of the corresponding node degrees. Node degree represents the significance of the associated input variable in QUBO formulation computation, gauging the extent of its involvement in nonzero scalar multiplications. Pruning is initiated from nodes with fewer connections.

**Step 2.** Row compression of $Q$ matrix is carried out in the order of sorted input vector list obtained in STEP 1. For each selected input variable $x_i$ from the list, if the respective row in matrix $Q$ is compressible, every nonzero element within the row is added to the element at its corresponding diagonal position, capitalizing on the symmetrical nature of $\mathbf{x}^T Q \mathbf{x}$ formulation. Subsequently, the elements in the compressed row are set to zero, with rows containing the updated diagonal elements marked as incompressible to ensure lossless compression.

**Step 3.** Column compression mirrors the operation conducted in STEP 2, with nonzero elements within the compressed column added to the their respective diagonal elements, then set to zero. Columns containing updated diagonal elements are labeled as incompressible.

**Step 4.** The QUBO matrix's compressed rows and columns, along with their corresponding variables in the input vectors, are eliminated, yielding the compressed QUBO matrix along with the compressed input vectors.

A detailed description of the compression methodology is featured in Fig. S6 in Sec. 4 of Supplementary Information. As a result, redundant rows and columns of QUBO matrix are removed, reducing crossbar array size required to implement the $\mathbf{x}^T Q \mathbf{x}$ QUBO formulation without sacrificing accuracy. The compressed QUBO matrix is mapped onto the FeFET crossbar for the iterative annealing of QUBO formalized COPs. The binary variable vectors $\mathbf{x_h}$ and $\mathbf{x_v}$ associated with the compressed QUBO formulation are applied to the WLs and SLs of the FeFET CiM array, respectively. In this way, the proposed compression approach enhances the scalability of CiM hardware, thereby scaling up the capacity for solving larger-scale COPs.

The efficacy of the proposed compression technique has been evaluated. For the same problem instances of graph coloring and Max-Cut COPs, as analyzed in Fig. 3a, b, respectively, the corresponding chip size reduction percentages are elucidated in Fig. 3e, f. These results demonstrate that the compression method yields substantial savings in chip size compared to the implementations without compression[38,39]. Note that the extent of chip size reduction does not necessarily correlates with the node count in a problem. This is because that the distribution of nonzero elements within the QUBO matrix significantly affects the impact of the compression method. For instance, if all nonzero elements aggregate within a single row, the QUBO matrix could be compressed to just one row, yielding high chip size savings. Conversely, if each row contains only one nonzero element, and these nonzero elements are at different columns, compression of the QUBO matrix might not be feasible, even if it is sparse.

### Solving COPs with multi-epoch simulated annealing

Previously developed FeFET CiM array demonstrates its capability to accelerate the computation of the QUBO formulation, which matches with annealing algorithms for solving COPs. That is, the configurations or solutions corresponding to the minimal QUBO energy are sought via an iterative annealing procedure. Simulated annealing (SA) algorithms were introduced to address the problem of local minimum trapping during the annealing process. The energy of the objective function, as computed by the configurations in current iteration, is compared with the energy state corresponding to current solution. If the computed energy is lower, the solution configurations are updated with the corresponding variable configurations. Conversely, if the energy is higher, the update retains a probability proportional to the temperature. Such annealing process has been adopted in prior NVM based annealers[38,39]. Nonetheless, conventional SA has demonstrated suboptimal performance in handling large-scale COPs[42]. To accelerate the SA process while still ensuring high probability of finding optimal solutions, a multi-epoch simulated annealing (MESA) algorithm is herein proposed.

Figure 4a illustrates the detail of the MESA process. For each epoch, an optimal solution $(\mathbf{x}_{hopt}, \mathbf{x}_{vopt})$ and its associated QUBO energy $E_{opt}$ are defined and sustained throughout the epoch. This records the lowest energy state attainable by the system, given the input configuration and the energy initialized to the optimal solution from the previous epoch. The QUBO energy, $E_{new}$, is calculated using the CiM hardware, as previously detailed. If $E_{new}$ is lower than the energy $E_o$ of the last iteration, indicating a progression toward a lower energy landscape, this QUBO energy and its associated variable solution $(\mathbf{x_h}, \mathbf{x_v})$ are accepted. The optimal solution $(\mathbf{x}_{hopt}, \mathbf{x}_{vopt})$ along with its energy value $E_{opt}$ within this epoch are either updated or maintained, depending on the comparison between $E_{new}$ and the optimal solution. Should $E_{new}$ closely approximate $E_o$, indicating that the system is trapped at a local minimum, the energy and its corresponding variable solution remain unaltered, and the trap count is updated. If $E_{new}$ is notably larger than the energy $E_o$, the system has a probability closely related to the temperature $T$ to accept the variable solution, allowing a chance to escape from local minimum. Subsequently, the system introduces random perturbations by flipping a few bits in the input vectors, and proceeds to the next iteration. When the system is trapped at a local minimum, where the energy trajectory remains stagnant for a predefined period $Count_{max}$, the epoch concludes, the temperature resets, and a fresh epoch commences. As a result, the length of each epoch is dynamically adjusted according to the system's advancement. The input configuration and its corresponding energy at the beginning of a new SA epoch are initialized based on the optimal solution and the energy recorded during the last epoch. Such initialization ensures the continuous convergence of MESA toward lower energy states.

Figure 4b shows the COP solving capability of MESA over that of conventional SA in identifying optimal solutions for prevalent Max-Cut problems. It can be seen that MESA outperforms the conventional SA across all Max-Cut problems, ranging in size from 800 to 3000 nodes, encompassing a search space spanning $2^{800}$ to $2^{3000}$. As the node count increases, i.e., the problem complexity grows, MESA yields superior solutions to SA, boasting an improvement of up to 27% in cut value when addressing 3000-node problems. With the dataset tested, Fig. S7 shows that MESA consistently outperforms conventional SA in terms of success rate and time-to-solution, achieving nearly double the speedup compared to conventional SA. The improvements of MESA over conventional SA result from the multi-epoch annealing process, where each annealing epoch begins with the optimal input configuration from the previous one. This greedy initialization enhances the convergence of the objective function towards the global optimum. Moreover, MESA includes an adaptive annealing termination scheme within each epoch to prevent the algorithm from trapping at a local minimum for an extended number of iterations, thus reducing unnecessary time and resource consumption. Leveraging both software advancements (introduction of MESA) and hardware improvements, particularly notable enhancements in hardware resource utilization,

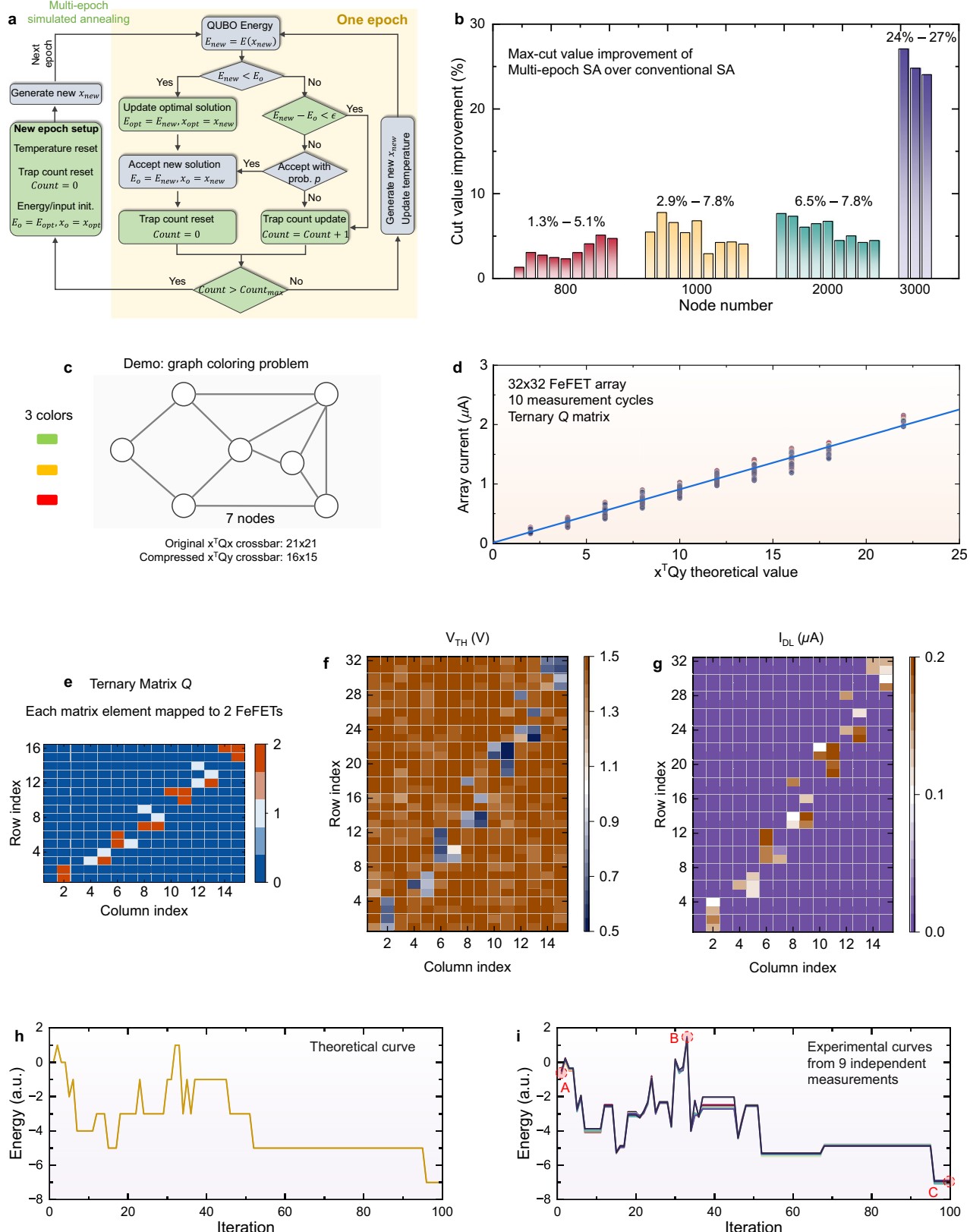

time efficiency, and energy efficiency associated with QUBO computations within each annealing iteration (Fig. 3e, f), our approach markedly achieves significant improvements across various aspects, including problem-solving scale, time efficiency, and solution quality. Yet, the capability of all the CiM SA solver relies not only on the algorithm, but also the precision of the hardware especially when

mapping the QUBO matrix. Figure S8 demonstrates the impact of the QUBO matrix precision in solving the prime factorization problem (PFP), as an example. A single MESA epoch of solving the PFP is studied. As expected, the higher the precision of QUBO matrix, the higher the success rates of MESA in finding the solution. This is because higher mapping precision yields a more accurate representation of the

**Fig. 4 | Demonstration of a new simulated annealing algorithm and its hardware acceleration using FeFET CiM array. a** Proposed multi-epoch simulated annealing (MESA) algorithm for solving complex COPs. **b** Results of 30 Max-Cut problems demonstrate that MESA outperforms conventional SA in the quality of solution. **c** A toy example of graph coloring for hardware implementation, which consists of 7 nodes and 3 colors to paint. After compression, it requires a ternary CiM array of 16 × 15 to map. **d** When mapped to a FeFET CiM array, the measured array current is linearly proportional to the theoretical $\mathbf{x}^T Q \mathbf{y}$ value, demonstrating the feasibility of performing MESA in hardware. **e** The ternary matrix $Q$ corresponding to the toy graph coloring problem in (**c**). **f** Using 2 FeFETs to represent each ternary $Q$ matrix element, the programmed FeFET $V_{TH}$ map of the corresponding 32 × 15 array. **g** The corresponding cell current. **h** The theoretical energy as a function of annealing iteration. **i** Experimentally measured energy as a function of annealing iteration, showing successful operation of the hardware in implementing the annealing algorithm.

## Table 1 | Summary of QUBO Solvers

| Reference | 41 | 50 | 51 | 52 | 42 | This work |
|---|---|---|---|---|---|---|
| Problem | Max-Cut | Max-Cut | Spin Glass | Graph Partion | Traveling Salesman | Graph Coloring/Max-Cut/Prime Factorization |
| QUBO matrix compression | No | No | No | No | No | Yes |
| method | SA | Chaotic SA | SA | SA | Multi-step SA | MESA |
| Hardware implementation | memristor based crossbar | memristor based crossbar | RRAM based crossbar | RRAM based crossbar | RRAM based crossbar | FeFET based crossbar† |
| Hardware acceleration* | VM | VM | VM | VM | VM | VMV |
| Hardware size | 60 × 60 | 2 × 2 | 11 × 3 | 64 × 64 | 1024 × 1152 | 32 × 32 |
| Problem size | 60 node | 5 node | 15 node | 6 node | 100 node | 21 node |

⋆: VM denotes vector-matrix multiplication, VMV denots vector-matrix-vector multiplication.
†: The first one known to us.

energy landscape, as shown in Fig. S9, which however incurs more hardware costs in terms of analog-digital-converter and array size.

To demonstrate the capability of developed FeFET CiM array in accelerating COPs, a toy example of graph coloring, as shown in Fig. 4c, is evaluated. To fit the problem into the developed 32 × 32 FeFET array, the example consists 7 nodes and 3 colors to be assigned. The initial QUBO formulation prior to compression necessitates an 21 × 21 array (i.e., each node can be any of the three color, thus total 21 input variables, per the QUBO conversion described in Sec. 1 of Supplementary Information), whereas the compressed formulation can be implemented on an 16 × 15 array, resulting in a notable 1.84 × reduction in chip size. In this example, the matrix $Q$ is ternary (i.e., has value of 0, 1, 2). For experimental demonstration, 2 FeFETs are used to represent each matrix element. Three FeFET CiM array dies (see Fig. 2c for the chip photo) have been employed to evaluate the QUBO formulation. The capability of the chip in realizing the intended $\mathbf{x}^T Q \mathbf{y}$ computation is demonstrated in Fig. 4d, where the measured array total current shows a linear dependence on the theoretical $\mathbf{x}^T Q \mathbf{y}$ value. Building upon this capability, MESA is performed on the chip. Figure 4e shows the ternary $Q$ matrix corresponding to the graph coloring problem in Fig. 4c. The corresponding 32 × 15 FeFET array is then programmed. Figure 4f, g show the $V_{TH}$ and measured cell current, respectively, demonstrating successful mapping of the matrix $Q$.

Figure 4h and i show the theoretical energy and experimentally measured energy evolution with annealing iterations. The experimental measurements are conducted on 3 separate dies. Figure 4i shows 9 separate measurements on one of the die, where for each measurement, the FeFET CiM array is erased and programmed again with the same QUBO matrix, and MESA is executed. All of the measured curves consistently align with the theoretical curve, validating the capability and robustness of the proposed FeFET CiM systems in performing MESA to solve COPs. More experimental measurement results can be found in Fig. S10. Figure S11 showcases the graph coloring configurations during annealing, highlighted at different iteration steps shown in Fig. 4i, i.e., the beginning (A), midpoint (B), and end (C) of the evolution process. Initially, errors such as multiple colors attributed to a single node, identical colors assigned to adjacent nodes, or uncolored nodes could occur, which correspond to high QUBO energy states like point A and B. In these cases, the algorithm is still exploring the solution space with the constraints loosely enforced. As annealing proceeds, the proposed approach can ultimately find the optimal solution with all the constraints satisfied.

## Discussion

We proposed a comprehensive hardware-algorithm co-design framework for solving the complex COPs efficiently. The proposed approach comprises a FeFET-based CiM array that accelerates the critical in-situ VMV multiplications within the QUBO formulation. Additionally, the proposed QUBO matrix compression technique significantly improves the chip utilization, thereby enhancing the problem solving capability of the hardware when addressing larger COPs. Complementing this, our multi-epoch based SA algorithm optimizes the proposed solver's ability to converge and reach optimal solutions within a shorter time period. Both the simulation and experimental measurements on fabricated prototypes validate the problem-solving capability of the proposed approach. The solver summary in Table 1 demonstrates that the proposed framework can outperform solvers for COPs commonly showcased in prior works by leveraging the VMV acceleration and QUBO compression techniques. Remarkably, the proposed framework showcases robust COP-solving capability and exhibits wide applicability to a broad spectrum of COPS that can be transformed to QUBO formulation. Importantly, our framework has the potential for broad adoption across various NVM based crossbars beyond FeFET devices. Moreover, our framework can accommodate various types of COPs with even larger scales than the size of FeFET CiM chip, as illustrated in Sec. 1 and Sec. 8 of Supplementary Information. This adaptability positions our approach as a universal and highly efficient method for QUBO computation and solving COPs using three-terminal voltage-driven structures.

## Methods

### FeFET chip integration

Testing chip is designed with FeFETs integrated on 28nm high-$\kappa$ metal gate (HKMG) platform. The fabricated ferroelectric field effect transistor (FeFET) features a poly-crystalline Si/TiN (2 nm)/doped HfO$_2$ (8 nm)/SiO$_2$ (1 nm)/p-Si gate stack. The ferroelectric gate stack process module starts with growth of a thin SiO$_2$ based interfacial layer, followed by the deposition of an 8 nm thick doped HfO$_2$. A TiN metal gate electrode was deposited using physical vapor deposition (PVD), on top

of which the poly-Si gate electrode is deposited. The source and drain n+ regions were obtained by phosphorous ion implantation, which were then activated by a rapid thermal annealing (RTA) at approximately 1000°C. This step also results in the formation of the ferroelectric orthorhombic phase within the doped $HfO_2$.

### FeFET chip electrical characterization

The measurements primarily utilize a PXIe measurement system provided by National Instruments. A padring comprising 28 individual analog and digital pads establishes connections between the 1kb (32 × 32) FeFET macro and a serial peripheral interface (SPI). The adapter board interfaces with specific pads on the wafer using a probecard within a wafer probe system. The setup includes distinct NI PXIe-4143 source measure units (SMU) and an NI PXIe-6570 pattern generator. Notably, the output pins of the latter device can function as a Pin Parametric Measurement Unit (PPMU). This configuration facilitates the generation of necessary supply, bias voltages, and digital signals. Moreover, the pattern generator plays a crucial role in configuring the scan chain for the proper addressing of wordlines and sourceline/drainline.

### Data availability

The data that support the findings of this study are available in Zenodo with the https://doi.org/10.5281/zenodo.10697395. [https://doi.org/10.5281/zenodo.10697394]. Other data related to this study are available from the corresponding author.

### Code availability

The code that support the findings of this study are available in Zenodo with the https://doi.org/10.5281/zenodo.10697395. [https://doi.org/10.5281/zenodo.10697394].

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

## Acknowledgements

No external funding is received to support this research.

## Author contributions

X.Y., T.K. and K.N. proposed and supervised the project. M.G., A.V., F.M., N.L. and T.K. designed the chip and performed the experimental verification of the proposed design. Y.Q., Z.S. and C.Z. conducted SPICE simulations and verification. Z.Z., Z.J., Y.S. and X.G. helped with data analysis. All authors contributed to write up of the manuscript.

## Competing interests

The authors declare no competing interests.
