## [Peer Review File · Nature Communications]

REVIEWER COMMENTS

Reviewer #1 (Remarks to the Author):

The manuscript, "First demonstration of ferroelectric compute-in-memory annealer for combinatorial optimization problems", reported the first integrated FeFET based compute-in-memory (CiM) annealer towards solving larger-scale combinatorial optimization problems (COPs). One good metric of this work is the inclusion of an FeFET CiM array chip (for in-situ vector-matrix-vector (VMV) multiplication) that is rarely seen in existing FeFET-based CiM designs. This work further makes use of this VMV operation function to support QUBO with asymmetric input vectors, and an annealing algorithm. As far as I know, this is the first experimental prototype of FeFET CiM chip for accelerating QUBO computations and possibly also solving complex COPs. From this point of view, this work is novel and interesting.

Below are some detailed comments:

1. There have already been some FeFET-based CiM works for matrix-vector multiplication, including those in the current domain and the charge domain. What is the key difference between the proposed VMV function with these works, in particular, from the array structure's perspective? In addition, some clarifications could be helpful for highlighting the contributions of the manuscript. For example, what is the difference from the 1T1R design in [38] for solving Max-cut problems? Would there be specific dependence on using FeFET rather than other devices?
2. It would be better to further explain how the proposed MESA leads to the improvement in Figure 4. Some supplementary experimental results could be added to the manuscript.
3. This work shows experimental results plus simulation results in solving COPs. My concern is that some of the problem scale could be much larger than the presented demonstration. How would the device non-ideality fit with very large-scale problems? In addition, as several COP problems are discussed, how would this proposed chip fit with other COPs?
4. Figure 2 labels are confusing, please try to make them clear. Also, some label text is overlapping.

Reviewer #2 (Remarks to the Author):

This work reports of the design and fabrication of ferroelectric FET based crossbar arrays for vector-matrix-vector multiplication, which can be used for hardware base optimization (such as simulated annealing). This direction is important and interesting because the optimization can be implanted in to the hardware directly and then fasten the optimization dramatically. Regarding this specific manuscript, I have several comments.

1. Many other elemental devices can also be used for VMV multiplication, such as floating gate FET, oxide based memristor. Regarding using ferroelectric FET, what is the advantage and disadvantage? The comparison should be added and discussed.
2. In Figure 1, many illustration have been presented. However, it is still not that clear to me about the working principle and advantage of the design. I would suggest the authors to present that in a more straightforward way.
3. More measurement data should be shown at individual FET level, such as time response.
4. Regarding the efficiency/performance improvement over conventional optimization, it is not that significant from the Figure 4b. I would suggest the authors to evaluate and compare the improvement in more details and to show there can be a huge improvement.

We thank the reviewers for their critical evaluation of our manuscript and constructive comments to further improve this work. We have substantially edited the paper based on the reviewers' feedback. Point-by-point response is attached here.

Reviewer 1

1. There have already been some FeFET-based CiM works for matrix-vector multiplication, including those in the current domain and the charge domain. What is the key difference between the proposed VMV function with these works, in particular, from the array structure's perspective? In addition, some clarifications could be helpful for highlighting the contributions of the manuscript. For example, what is the difference from the 1T1R design in [38] for solving Max-cut problems? Would there be specific dependence on using FeFET rather than other devices?

Figure R1: *Comparison of FeFET-based CiM for vector-matrix multiplication (VMM) and vector-matrix-vector (VMV) multiplication. a. FeFET-based CiM for VMM. b. FeFET-based CiM for VMV multiplication. c. Traditional CiM approaches supporting VMM acceleration require two steps to realize $\vec{x}^T Q \vec{y}$, d. while our proposed method performs $\vec{x}^T Q \vec{y}$ in a single step.*

We appreciate the reviewer's insightful observation. In this work, we have been leveraging the current-domain CiM by analog summation of the column current. We have adopted the 1FeFET1R structure to suppress the current variation resulted from the FeFET device variation. Our contributions are not proposing a fundamentally dif-

ferent analog summation scheme. In our proposed work, the unique three-terminal structure and the inherent single transistor 3-input AND logic of the FeFET device has been fully exploited to implement vector-matrix-vector (VMV) function in a single step, demonstrating a significantly higher level of efficiency.

This single-step VMV operation is advantageous compared with conventional vector-matrix-multiplication (VMM) operation, especially for the combinatorial optimization problems (COPs), where the objective function is of concern. Fig. R1(a) and (b) illustrate VMM and VMV multiplication functions from the array structure’s perspective. In previous FeFET based CiM implementations for VMM, the source lines (SLs) of the FeFET based crossbar are all set to high-level voltage, where the column currents represent the vector output of $\vec{x}^T Q$. In our proposed CiM array for VMV multiplication, SLs are set according to the input vector \vec{y} , with the summed current from the crossbar columns directly representing the output of $\vec{x}^T Q \vec{y}$. This difference results in much improved efficiency of the proposed VMV scheme in accelerating the core operation of QUBO. Fig. R1(c) and (d) illustrate the conceptual realizations of the core operation of QUBO based on prior works and our approach in (a) and (b), respectively. In previous works, after VMM and analog-to-digital (A/D) conversion, the output result of VMM, i.e., $\vec{x}^T Q$, is dot multiplied with \vec{y} to obtain the output value of $\vec{x}^T Q \vec{y}$. This process involves computations in both analog and digital domains, necessitating energy and time consuming data movement and conversion. In contrast, our proposed design directly computes $\vec{x}^T Q \vec{y}$ in the analog domain in a single step, followed by an A/D conversion. Our approach eliminates the need for dot product implementations, significantly reducing the need for data transfer and making the overall process more energy and time efficient.

Another key difference lies in the matrix compression technique enabled by our approach. Our proposed lossless QUBO compression transforms the general QUBO form

$\vec{x}^T Q \vec{x}$ to a compressed form $\vec{x}_h^T Q' \vec{x}_v$ with two asymmetric input vectors, resulting in a notable reduction in matrix and input vector sizes. This leads to substantial chip size savings (up to 75%, as depicted in Fig. 3(f)) or enhances the problem-solving capacity for larger-scale COPs. In contrast, prior works supporting VMM cannot adopt the matrix compression method, limiting problem-solving capacity.

In comparison to the work presented in [38], our proposed approach offers several contributions: (i) Unlike [38], which employs a 1T1R structure, our work utilizes a more compact 1FeFET1R structure (due to its voltage-driven write mechanism), naturally performing single-step VMV operation. Furthermore, compared to current-driven ReRAM, the voltage-driven programming of FeFET results in significantly lower energy consumption. These advantages of 1FeFET1R structure ensures both the compactness and the computation efficiency compared to the design in [38]. (ii) While [38] utilizes the 1T1R array for VMV multiplication with identical inputs, our work considers a more general and efficient VMV multiplication with asymmetric inputs. Such VMV multiplication acceleration with two asymmetric inputs enables highly efficient QUBO matrix compression, resulting in significant chip size savings (up to 75%, as depicted in Fig. 3f), and more efficient QUBO computations. (iii) At algorithm level, [38] employs a traditional simulated annealing (SA) algorithm. In our work, we propose a multi-epoch SA (MESA), which achieves up to a 27% improvement in solution value compared to SA when solving Max-cut problems, as demonstrated in Fig. 4b. (iv) More importantly, our work provides experimental validation of solving COPs through a fabricated 1FeFET1R chip. In contrast, [38] solely presents a proof of concept at the simulation level for solving the Max-Cut problem.

Compared to other NVM devices, like PCM, ReRAM, FeFET stands out for its ability to support in-situ VMV multiplication, thanks to its three-terminal structure, voltage-driven mechanism and inherent single transistor AND logic. However, other

NVM based designs with three terminals, such as 1T1R cell in [38], can also achieve VMV multiplication by applying two input voltages at the gate of the access transistor and one terminal of the ReRAM device, respectively. The ReRAM stores the QUBO coefficient, and the current flowing through the 1T1R structure represents the output of VMV multiplication. Therefore, our proposed VMV multiplication technique, along with the lossless QUBO compression method, holds the potential for adoption in a broad spectrum of NVM based crossbars. This versatility positions our approach as a general method for highly efficient QUBO computation and COP solving.

We have added clarifications regarding the differences between our proposed approach and the 1T1R design [38], and the universality and efficiency of our proposed framework in using other devices beyond FeFETs for COP solving in the last paragraph of Section "Introduction", and Section "Conclusion", respectively, to further highlight the contributions of our manuscript.

2. It would be better to further explain how the proposed MESA leads to the improvement in Figure 4. Some supplementary experimental results could be added to the manuscript.

We thank the reviewer for the suggestion. The proposed MESA introduces a distinctive approach compared to traditional SA which employs a single annealing process. MESA comprises multiple epochs, each conducting an individual annealing process. Within each epoch, we initiate the input configuration as the one corresponding to the optimal object function value from the previous epoch. This greedy initialization enhances the convergence of the objective function, resulting in more efficient problem-solving. Moreover, MESA incorporates a dynamic annealing termination scheme within each epoch. If the objective function remains unchanged for a certain number of iterations, MESA promptly concludes the current epoch and transitions to the next one. This adaptive mechanism prevents the algorithm from trapping at a

local minimum for an extended number of iterations, thereby mitigating unnecessary time and resource consumption.

Figure R2: *Energy evolution of MESA and SA in solving a Max-Cut problem with 800 nodes.*

In response to the reviewer’s suggestion, we have added the explanation in the third paragraph of Section ‘Solving COPs with multi-epoch simulated annealing’ to clarify the improvements offered by MESA over traditional SA. Additionally, we have included a supplementary figure as Fig. S7b (shown as Fig.R2 here in the response) and corresponding text that illustrate the energy evolution of MESA and SA when solving a Max-Cut problem.

3. This work shows experimental results plus simulation results in solving COPs. My concern is that some of the problem scale could be much larger than the presented demonstration. How would the device non-ideality fit with very large-scale problems? In addition, as several COP problems are discussed, how would this proposed chip fit with other COPs?

Figure R3: *Large-scale COPs solved by multi FeFET chips.* **a.** Large-scale COPs. **b.** COPs converted to general QUBO form $\vec{x}^T Q \vec{x}$. **c.** $\vec{x}^T Q \vec{x}$ losslessly compressed into $\vec{x}_h^T Q' \vec{x}_v$. **d.** Q' being decomposed into multiple sub-matrices. **e.** $\vec{x}_h^T Q' \vec{x}_v$ being decomposed into multiple forms. **f.** Multiple forms being mapped to multiple chips for $\vec{x}_h^T Q' \vec{x}_v$ computations.

We thank the reviewer for raising this concern. We believe that the scalability of our work is exactly a significant advantage compared with many COP solvers, especially those based on dynamical Ising Machines. In our design, when addressing larger-scale COPs that surpass the capacity of our chip, we employ a decomposition and solution strategy involving multiple FeFET chips, as illustrated in Fig. R3. Initially, the large-scale COP (a) is converted to QUBO form, and then (b) losslessly compressed into (c) a more compact QUBO form $\vec{x}_h^T Q' \vec{x}_v$. This compact form (d) is then decomposed into several smaller forms (e), each of which is (f) mapped to individual FeFET chips for separate QUBO computations. The final QUBO value, corresponding to the original objective function of large-scale COP, results from combining the outputs of the decomposed forms processed on multiple chips. Or equivalently, these decom-

posed smaller forms can be carried out in the time domain using a single FeFET chip. In this approach, the smaller matrices are sequentially programmed into the FeFET chip, and the calculated results are temporarily buffered and summed together at the end. This decomposition process indicates that our proposed framework efficiently handle larger-scale COPs. In this way, our chip can accommodate larger-scale COPs effectively while maintaining good linearity and mitigating the impact of device non-ideality. This scalability represents a significant advantage of our approach compared to other Ising machines that struggle with scalability and problem decomposition for larger-scale COPs.

Regarding different COPs, in our supplementary materials, specifically in Section 1 'Conversions from COPs to QUBO formulation,' we have presented a few COPs, including Max-Cut (Supplementary 1.1), graph coloring (Supplementary 1.2), and prime factorization (Supplementary 1.3). Each COP is thoroughly described how they can be converted into QUBO forms. After the conversions, our proposed chip can accelerate the QUBO computations and identify the optimal input variable values. These values can then be translated into the corresponding solutions for these COPs. Therefore, the proposed framework is intrinsically a QUBO accelerator. As long as a problem can be converted into a QUBO form, it could be accelerated and solved by our approach.

We have added how we deal with larger-scale COPs that surpass the capacity of our chips in Section 8 "Addressing larger-scale COPs" of Supplementary Information, and that our proposed chip fits with other COPs in Section "Conclusion".

4. Figure 2 labels are confusing, please try to make them clear. Also, some label text is overlapping.

We appreciate the reviewer's suggestion. We have revised the labels of Fig. 2(e) and (h) to make them clear, where label text is no longer overlapping.

Reviewer 2

1. Many other elemental devices can also be used for VMV multiplication, such as floating gate FET, oxide based memristor. Regarding using ferroelectric FET, what is the advantage and disadvantage? The comparison should be added and discussed.

Figure R4: *NVM-based VMV multiplication. a. Floating gate FET array. b. Memristor array.*

Thanks for your question and suggestion. As depicted in Fig. R4, various other devices, such as floating gate FET and two-terminal resistive memory (e.g., ReRAM, PCM, etc.) with a transistor selector can be employed for VMV multiplication, as they share the characteristic of having three-terminals to apply the two inputs vectors.

That said, FeFET stands out due to its CMOS compatibility and its scalability beyond 28nm [1]. In contrast, flash struggles to scale below the 28nm node [2]. When performing VMV multiplication, a FeFET based CiM array necessitates lower write/read

voltages and less write time compared to flash [3]. This results in reduced energy consumption and execution time. Moreover, FeFET features a voltage-driven write and read mechanism and a substantial ON/OFF ratio, simplifying the required sensing circuitry. In contrast, current-driven memristor devices require additional access transistors, leading to much more energy consumption than FeFETs. Nevertheless, it's worth noting that the technology of FeFET is still under active research and development, not as mature as that of flash and memristors. The largest fabricated FeFET CiM chip, as demonstrated in our work, has a size of 32×32 (1KB), while RRAM based chips can achieve sizes up to 4MB [4]. Additionally, the endurance of Si FeFETs, approximately in $10^5 - 10^9$ range, still needs improvement, whereas memristors like PCM and RRAM can achieve $> 10^9$ and $10^6 - 10^{12}$, respectively [5]. However, new FeFET designs have kept emerging with recent impressive endurance being demonstrated in back-end-of-line FeFET [6, 7] and Si FeFET with interface engineering [8]. More discussions on the advantages and disadvantages of using FeFET can be found in [9, 10].

We have added the comparisons and related discussions between FeFET and floating gate FET, memristor in the last paragraph of Section "Introduction".

2. In Figure 1, many illustration have been presented. However, it is still not that clear to me about the working principle and advantage of the design. I would suggest the authors to present that in a more straightforward way.

Thank you for your suggestion. Fig.R5 depicts the flow of our proposed framework, and we illustrate its working principle as below: (i) A COP is initially converted into a QUBO formulation $\vec{x}^T Q \vec{x}$. (ii) This QUBO formulation is then losslessly compressed into a more general and compact form with asymmetric variable vectors $\vec{x}_h^T Q' \vec{x}_v$. (iii) The QUBO matrix Q' of the compressed formulation is mapped onto a FeFET based crossbar array, which inherently performs single-step VMV multiplication. The summed current of the crossbar represents the value of the compressed QUBO for-

mulation. (iv) The solving process utilizes a multi-epoch simulated annealing (MESA) algorithm. In each iteration, the FeFET based array computes the QUBO formulation value, and the objective function value is determined. (v) After the MESA process, the variable vector configurations that correspond to the optimal objective function value are obtained and translated into the solution for the given COP.

Figure R5: *Flow of our proposed framework. a.* Various types of COPs are converted to **b.** QUBO form $\vec{x}^T Q \vec{x}$, which is **c.** losslessly compressed into more general and compact $\vec{x}_h^T Q' \vec{x}_v$. **d.** $\vec{x}_h^T Q' \vec{x}_v$ is mapped to FeFET-based crossbar for QUBO computation, **e.** within each iteration of MESA process. **f.** The solutions of the COPs are ultimately obtained.

Our CiM based annealer could address the challenges faced by previous digital annealers and dynamical Ising solvers, offering several advantages: (i) Our CiM based approach stores the QUBO coefficient matrix in memory and directly performs VMV

multiplications in-memory, avoiding the data movement bottleneck seen in digital annealers; (ii) Our proposed CiM annealer accurately maps coupling matrix coefficients by programming multiple FeFET devices with binary states to represent a single matrix coefficient value. In contrast, dynamic Ising solvers use coupling implementations between spins with multi-level devices, which are often sensitive to noise and programming deviations; (iii) Our CiM approach easily handles larger-scale problems beyond the capacity of our chip by decomposing the corresponding QUBO formulation into smaller forms, then independently mapping and computing these forms across our CiM chips. In contrast, dynamic Ising solvers struggle to decompose the larger-scale problems and distribute them across multiple solvers. This is due to the stringent requirement for dynamic synergy between Ising solvers, as they may evolve at different rates; (iv) Our CiM array can readily implement self-interaction terms within the QUBO formulation by programming the diagonal matrix coefficient values onto the crossbar cells. In contrast, dynamic Ising solvers are constrained to handling COPs without self-interaction terms because their spin-to-spin connections cannot support self-feedback. As a result, dynamic Ising solvers are primarily applicable to a specific subclass of COPs that do not involve self-interaction terms.

In response to the reviewer's suggestion, we have added corresponding descriptions in the last paragraph of Section "Introduction" and Section 2 "Flow of proposed framework" of Supplementary Information to illustrate the working principle of our design in a more straightforward way, and modified Fig.1 by adding more explanatory words into Fig.1f and g. We have revised corresponding text in the second last paragraph of Section "Introduction" to emphasize the advantages of our design over other COP solvers.

3. More measurement data should be shown at individual FET level, such as time response.

Thanks for your suggestion. In response to this suggestion, we have incorporated the switching dynamics of an individual FeFET, which has the same dimensions as the FeFETs integrated into the array ($0.45\mu\text{m}/0.45\mu\text{m}$). Fig.R6 now presents the measured switching dynamics for both the set and reset processes. These dynamics clearly illustrate the tradeoff between programming speed and amplitude, as demonstrated in the supplementary Fig.S5b for the array.

Figure R6: *Switching dynamics of a FeFET under different write pulse amplitudes and pulse width.* The color represents the memory window. The boundary line indicates a memory window of 1V. The left half figure represents the scenario that a FeFET is initialized to 4V every time and then a negative reset gate pulse is applied, following which the memory state is measured. Similarly for the right half figure, the device is initialized with -4V write pulse.

We have included the measured switching dynamics, representing the time response of the FeFET, into Section 3 “FeFET CiM chip integration & testing” of Supplementary Information.

4. Regarding the efficiency/performance improvement over conventional optimization, it is not that significant from the Figure 4b. I would suggest the authors to evaluate and compare the improvement in more details and to show there can be a

huge improvement.

Thanks for your suggestion. We would like to clarify the significant improvements our approach offers over conventional SA algorithm, as shown in Fig. 4b. The cut value in Max-Cut problems is the key metric for evaluating the quality of problem solvers, as the object of Max-Cut problems is to find a vertex partition in a graph such that the number of edges crossing the partition is maximized [11]. Our MESA demonstrates a remarkable up to 27% cut value improvement in solving 3000-node Max-Cut problems compared to traditional SA. Additionally, MESA consists of multiple epochs, each conducting individual annealing process. Within each epoch, we initiate the input configuration as the one corresponding to the optimal object function value from the previous epoch. Such greedy initialization and multiple annealing processes lead to more efficient convergence and greater capability in finding improved solutions. As a complement to the solution quality demonstrated in Fig. 4b, we also present the speed and success rate of our MESA algorithm in Fig. S7. MESA consumes 2x reduction in iteration number than conventional SA when solving the same number of problems. Moreover, when tackling the Stanford Max-Cut problem set Gset [12], and considering a solution found above 90% of the optimal max-cut value as successful, MESA demonstrates a higher success rate, reaching 100%, compared to conventional SA's 90%. This indicates that MESA is capable of addressing more complex problems where conventional SA may encounter difficulties. In summary, our proposed MESA algorithm achieves a 27% improvement in solution quality, a 2x speedup and a 100% success rate in solving Max-Cut problems compared to conventional SA algorithm. Note that this advantage of MESA is independent of the underlying hardware, meaning that the improvements in solution quality, solving speed and success rate can be applied to any other CiM based annealers.

Due to the inherent nature of implementing VMV multiplication, our proposed

FeFET based CiM array enables significantly faster QUBO computations within each annealing iteration, compared to conventional CiM implementing VMM operations. Furthermore, our CiM array can handle QUBO formulations that exceed the chip's scale by employing our proposed lossless compression technique. Evaluation results in Fig. 3(e) and (f) demonstrate that our proposed array can achieve a 75% reduction in chip utilization when solving Max-Cut problems compared to prior CiM arrays implementing VMM.

Combining the algorithmic and hardware level innovations, our approach substantially achieves a huge improvement in solving efficiency in terms of solving scale, solving time, solving rate and solution quality, etc.

Following the reviewer's suggestion, we have added Figure S7(b) along with Figure S7(a) in Section 5 "Example: MESA analysis over Max-Cut problem" to emphasize that our proposed MESA achieves nearly 2x speedup, 100% success rate besides the 27% solution quality improvement compared to conventional SA. We have revised the corresponding text in the third paragraph of Section "Solving COPs with multi-epoch simulated annealing" to provide detailed improvements offered by our hardware-software co-design approach.

References

- [1] Salahuddin, S., Ni, K. & Datta, S. The era of hyper-scaling in electronics. *Nature Electronics* **1**, 442–450 (2018).
- [2] Nebashi, R., Banno, N., Miyamura, M., Bai, X., Funahashi, K., Okamoto, K., Iguchi, N., Numata, H., Sugibayashi, T., Sakamoto, T. *et al.* A 171k-lut nonvolatile

- fpga using cu atom-switch technology in 28nm cmos. In *2020 30th International Conference on Field-Programmable Logic and Applications (FPL)*, 323–327 (IEEE, 2020).
- [3] Banerjee, W. Challenges and applications of emerging nonvolatile memory devices. *Electronics* **9**, 1029 (2020).
- [4] Lanza, M., Sebastian, A., Lu, W. D., Le Gallo, M., Chang, M.-F., Akinwande, D., Puglisi, F. M., Alshareef, H. N., Liu, M. & Roldan, J. B. Memristive technologies for data storage, computation, encryption, and radio-frequency communication. *Science* **376**, eabj9979 (2022).
- [5] Sk, M. R., Thunder, S., Müller, F., Laleni, N., Raffel, Y., Lederer, M., Pirro, L., Chohan, T., Hsuen, J.-H., Wu, T.-L. *et al.* 1f-1t array: Current limiting transistor cascaded fefet memory array for variation tolerant vector-matrix multiplication operation. *IEEE Transactions on Nanotechnology* (2023).
- [6] Sharma, A. A., Doyle, B., Yoo, H. J., Tung, I.-C., Kavalieros, J., Metz, M. V., Reshotko, M., Majhi, P., Brown-Heft, T., Chen, Y.-J. *et al.* High speed memory operation in channel-last, back-gated ferroelectric transistors. In *2020 IEEE International Electron Devices Meeting (IEDM)*, 18–5 (IEEE, 2020).
- [7] Dutta, S., Ye, H., Khandker, A. A., Kirtania, S. G., Khanna, A., Ni, K. & Datta, S. Logic compatible high-performance ferroelectric transistor memory. *IEEE Electron Device Letters* **43**, 382–385 (2022).
- [8] Tan, A. J., Liao, Y.-H., Wang, L.-C., Shanker, N., Bae, J.-H., Hu, C. & Salahuddin, S. Ferroelectric hfo₂ memory transistors with high- κ interfacial layer and write endurance exceeding 10^{10} cycles. *IEEE Electron Device Letters* **42**, 994–997 (2021).
- [9] Keshavarzi, A., Ni, K., Van Den Hoek, W., Datta, S. & Raychowdhury, A. Ferro-electronics for edge intelligence. *IEEE Micro* **40**, 33–48 (2020).

- [10] Kim, K.-H., Karpov, I., Olsson III, R. H. & Jariwala, D. Wurtzite and fluorite ferroelectric materials for electronic memory. *Nature Nanotechnology* 1–20 (2023).
- [11] Glover, F., Kochenberger, G. & Du, Y. A tutorial on formulating and using qubo models. *arXiv preprint arXiv:1811.11538* (2018).
- [12] Stanford Max-Cut dataset. <https://web.stanford.edu/~yyye/yyye/Gset/>.

REVIEWERS' COMMENTS

Reviewer #1 (Remarks to the Author):

The authors well address my concerns. I'm generally satisfied with the revision of this work. However, I suggest the authors carefully revise the writing and fix minor issues. Some examples are shown below.

- FeFETs stands out due to its CMOS compatibility and scalability → FeFETs stand out due to their CMOS compatibility and scalability
- beyond 28nm node → beyond the 28nm node
- a FeFET based CiM array → a FeFET-based CiM array (many similar cases in the manuscript)
- Compared to memristor based Max-Cut problem solver [38, 39] → Compared to the memristor-based Max-Cut problem solvers [38, 39]
- depicted in Sec.2 → is depicted in Sec.2

Reviewer #1 (Remarks on code availability):

There is no code that needs to be reviewed. However, readers who would like to repeat the work could ask the authors for code.

Reviewer #2 (Remarks to the Author):

The authors have explained the questions arising from the alst round review and revised the manuscript accordingly. This work is overall a good demonstration of FeFET based VMV devices. Here I have some further minor comments. First is that the quantitative discussion/comparison of the theoretical/ultramate perfromance between this approach and other VMV approaches should be added, in which we can see the adv/disadv of FeFET based VMV. Second is that the perfromance improvement of 27% mentioned in the Figure 4 is not significant. Much more improvement is needed to show the advancement.

We thank the reviewers for their critical evaluation of our manuscript and constructive comments to further improve this work. We have substantially edited the paper based on the reviewers' feedback. Point-by-point response is attached here.

Reviewer 1

1. The authors well address my concerns. I'm generally satisfied with the revision of this work. However, I suggest the authors carefully revise the writing and fix minor issues. Some examples are shown below.

- FeFETs stands out due to its CMOS compatibility and scalability -> FeFETs stand out due to their CMOS compatibility and scalability

- beyond 28nm node -> beyond the 28nm node

- a FeFET based CiM array -> a FeFET-based CiM array (many similar cases in the manuscript)

- Compared to memristor based Max-Cut problem solver [38, 39] -> Compared to the memristor-based Max-Cut problem solvers [38, 39]

- depicted in Sec.2 -> is depicted in Sec.2

Thank you for dedicating your time and efforts to reviewing our manuscript. We are pleased to hear that the revisions have met your satisfaction. According to your suggestions, we have carefully revised the writing and addressed several minor issues.

2. There is no code that needs to be reviewed. However, readers who would like to repeat the work could ask the authors for code.

Once again, thank you for your valuable input and continued support of our work.

Reviewer 2

1. The authors have explained the questions arising from the first round review and revised the manuscript accordingly. This work is overall a good demonstration of FeFET based VMV devices. Here I have some further minor comments. First is that the quantitative discussion/comparison of the theoretical/ultimate performance between this approach and other VMV approaches should be added, in which we can see the advantage/disadvantage of FeFET based VMV. Second is that the performance improvement of 27% mentioned in the Figure 4 is not significant. Much more improvement is needed to show the advancement.

We appreciate the time and effort you invested in reviewing our manuscript. According to your suggestions, we have added additional quantitative results in the last paragraph of Section "Introduction", which emphasizes the advantages of FeFET technology against other NVMs. Simultaneously, we are actively investigating the optimization of MESA, aiming to achieve greater improvements in our future work.